# Distinct Minor Splicing Patterns across Cancers

**DOI:** 10.3390/genes13020387

**Published:** 2022-02-21

**Authors:** Lauren Levesque, Nicole Salazar, Scott William Roy

**Affiliations:** Department of Biology, San Francisco State University, San Francisco, CA 94132, USA; llevesque@mail.sfsu.edu (L.L.); roy@sfsu.edu (S.W.R.)

**Keywords:** U12 splicing, minor introns, cancer, TCGA

## Abstract

In human cells, the U12 spliceosome, also known as the minor spliceosome, is responsible for the splicing of 0.5% of introns, while the major U2 spliceosome is responsible for the other 99.5%. While many studies have been done to characterize and understand splicing dysregulation in cancer, almost all of them have focused on U2 splicing and ignored U12 splicing, despite evidence suggesting minor splicing is involved in cell cycle regulation. In this study, we analyzed RNA-seq data from The Cancer Genome Atlas for 14 different cohorts to determine differential splicing of minor introns in tumor and adjacent normal tissue. We found that in some cohorts, such as breast cancer, there was a strong skew towards minor introns showing increased splicing in the tumor; in others, such as the renal chromophobe cell carcinoma cohort, the opposite pattern was found, with minor introns being much more likely to have decreased splicing in the tumor. Further analysis of gene expression did not reveal any candidate regulatory mechanisms that could cause these different minor splicing phenotypes between cohorts. Our data suggest context-dependent roles of the minor spliceosome in tumorigenesis and provides a foundation for further investigation of minor splicing in cancer, which could then serve as a basis for novel therapeutic strategies.

## 1. Introduction

RNA splicing is the process by which introns are excised from pre-mRNA and the exons are then ligated together [1,2]. It is an essential and tightly regulated process in the maturation of transcripts. Alternative splicing allows for multiple transcripts to be made from a single gene, leading to an increase in protein diversity as well as another layer of gene expression regulation. Alternative splicing is regulated in a tissue-specific, developmental stage-specific manner, or by external stimuli. For instance, the splicing factor, PTB1, plays a pivotal role in neuronal differentiation by promoting neuron-specific splicing events [3,4]. In addition, CLK1, a SR protein kinase, is necessary for proper mitosis and regulates alternative splicing in a cell cycle specific manner [5].

Dysregulation of splicing is a common feature of many cancers, and can be caused by alterations in the splicing machinery, pre-mRNA *cis*-elements, or regulatory splicing factors [1,6]. These splicing changes regulate many cellular transformation processes known as the hallmarks of cancer [7,8]. For example, pre-mRNAs undergo alternative splicing to acquire traits of high-grade malignancy during the epithelial–mesenchymal transition (EMT), an important transdifferentiation step in carcinoma progression [9]. While many studies have been done to characterize and understand splicing dysregulation in cancer, almost all of them have assumed it is due to changes in activity of the U2 spliceosome and have ignored the U12 spliceosome.

In human cells, the U12 spliceosome, also known as the minor spliceosome, is responsible for the splicing of 0.5% of introns, while the major U2 spliceosome is responsible for the other 99.5% [10,11]. The minor spliceosome is comprised of four specific snRNPs: U11, U12, U4atac, and U6atac; which are different but equivalent to the U2 snRNPs (U1, U2, U4, and U6, respectively). The minor spliceosome-specific snRNPs include a combination of minor spliceosome-specific proteins and proteins that are a part of both spliceosomes. In addition, the U5 snRNP, as well as other non-snRNP factors, are shared between the two spliceosomes [11]. In addition to differences in composition, the two machineries are also responsible for the splicing of different introns, where minor introns differ from major introns in their 5′ss and branch point sequences [10,11]. Minor intron-containing genes have been shown to be functionally enriched in DNA repair, DNA replication, and transcription [10,11]. In addition to this functional enrichment, a variety of experimental data suggest minor splicing is involved in cell cycle regulation.

Some evidence suggests that minor splicing could be tumor-promoting. Recently, mouse embryonic stem cells were shown to have high minor splicing activity, which promotes the expression of SR proteins (essential regulators of splicing), which then decreased through differentiation [12]. In addition, knockdown of the minor spliceosome component, SNRNP48, in HeLa cells not only led to inhibition of minor splicing but also decreased proliferation [13].

However, there is also evidence suggesting that minor splicing has a tumor suppressor effect. In hematopoietic cancers, loss of function mutations is common in the minor spliceosome component ZRSR2, suggesting it might act as a tumor suppressor [14,15]. In support of this hypothesis, a recent paper showed that knockout of ZRSR2 increased proliferation of hematopoietic stem cells, which is mediated through decreased minor splicing, particularly in the LZTR1 gene, a negative regulator of Ras GTPases [16]. Similarly, mutations in the minor spliceosome component, CENTATAC, (also known as CCDC84) led to impaired chromosome segregation, which correlated with inefficient minor splicing [17]. Furthermore, the tumor suppressor PTEN contains a minor intron, and its expression is positively regulated by the splicing of that minor intron [18].

This intriguing yet contradictory evidence suggests minor splicing could play a role in cancer progression; however, no one has systematically investigated this relationship. Recently, RNA-seq data from The Cancer Genome Atlas (TCGA) has been used to study alternative splicing in cancer, as identical preparation across tissue samples affords the opportunity for controlled comparison of spliced-to-unspliced ratios across samples [19,20,21,22,23]. In this study, we used TCGA RNA-seq data to analyze differences in minor splicing between tumor and adjacent normal tissue at the transcriptomic level across 14 different cohorts. Given previous evidence that regulation of overall minor spliceosomal activity is one mechanism of coordinated regulation of many minor intron-containing genes, we sought to test whether there was evidence for changes in the overall level of minor spliceosomal activity, as assessed by minor intron retention.

## 2. Materials and Methods

### 2.1. Data Sources

Exon junction alignment data for all TCGA samples was generated by Kahles et al., 2018 [19]. The data for all intron retention events was downloaded from https://gdc.cancer.gov/about-data/publications/PanCanAtlas-Splicing-2018 (accessed on 18 February 2021), with the file name as “merge_graphs_intron_retention_C2.counts.hdf5”. For gene expression analysis, gene counts data was accessed through the web-portal of TCGA for unique patient tumor samples and adjacent normal tissue for each cohort (https://portal.gdc.cancer.gov/ (accessed on 22 February 2021)). After filtering for samples of interest, counts were normalized using DESeq2 [24].

### 2.2. Minor Intron Retention Analysis across TCGA Cohorts

From the exon junction alignment data, introns were selected for those that were classified as minor introns in the IAOD (https://introndb.lerner.ccf.org/ (accessed on 18 February 2021)) [25]. Of the 673 minor introns annotated in the GRCh37 human genome, 110 minor introns were detected as intron retention events in the intron retention data set generated by Kahles et al., 2018 [19] using the tool SplAdder [26]. Analysis was performed for all TCGA cohorts that had at least 5 patients with matched tumor and normal samples. For each minor intron, PSI (percent spliced in) was calculated for all samples in the cohort that had at least 1 read spanning the spliced or retained minor intron. This was done using the following equation: COV_INT_/(COV_INT_ + COV_EJ_) × 100, where COV_INT_ is the mean coverage of the retained intron (total number of alignments to intron/length of intron), and COV_EJ_ is the number of spliced alignments spanning the intron. If there were at least 3 patients in the cohort that had PSI values for both tumor and normal tissue, differential splicing of the intron was analyzed using a paired Wilcoxon signed-rank test (FDR < 0.05).

### 2.3. Gene Expression Analysis across TCGA Cohorts

For the 623 genes that contained a minor intron (as annotated in the IAOD [21]), differential gene expression between tumor and normal tissue was analyzed using a paired Wilcoxon signed-rank test (FDR < 0.05) for each of the 14 TCGA cohorts. The same analysis was done for the other gene sets: protein-coding genes (19,634 genes, annotated in GENCODE [27]), minor spliceosome components (15 genes, literature curated), and stem cell related genes (48 genes, human adult stem cell gene signature from Smith et al. [28]).

### 2.4. Mutation of MIGs across TCGA Cohorts

The GDC Data Portal [https://portal.gdc.cancer.gov/ (accessed on 21 July 2021)] was used to identify minor intron-containing genes that are frequently mutated in cancer. From the 623 genes that contained a minor intron (as annotated in the IAOD [25]), 18 were identified as frequently mutated in cancer. For each of those genes, the percent of cases with a mutation was obtained from the GDC Data Portal across the 14 TCGA cohorts.

### 2.5. Estimation of Minor Splicing Activity in Tumors

Relative overall minor splicing activity was estimated for each tumor based on PSI for 110 minor introns. First, β_i_ coefficients were calculated based on PSI values using the following formula: β_i_ = (μ_i_ − T_i_)/σ_i_ (i = 1,2, …, n), where μ_i_ and σ_i_ represent the mean and standard deviation of the ith intron’s PSI across all normal tissue samples, respectively; n is the total number of introns, and T_i_ is the ith intron’s PSI in a tumor sample. Next, the summation of βi coefficients across all 110 introns were summed up to estimate minor splicing activity of the tumor sample.

### 2.6. Estimation of Changes in Minor Splicing Activity between Matched Tumor and Normal Tissues

Changes in minor splicing activity were estimated for each patient based on changes in PSI for 110 minor introns between matched tumor and adjacent normal tissue. For each minor intron, dPSI values were calculated (normal−tumor). Next, the dPSI values across all 110 introns were summed up to estimated changes in minor splicing activity in the patient.

### 2.7. Functional Enrichment Analysis

The bioinformatic tool, PANTHER [29], was used to test for significant functional enrichment in GO and Reactome terms (FDR < 0.05), using the 623 minor intron-containing genes as the reference list.

### 2.8. Correlation of Changes in Gene Expression and Minor Splicing across TCGA Cohorts

Each TCGA cohort was given a minor splicing score, which was calculated using the following equation: INT_up_−INT_down_, where INT_up_ was the number of minor introns that showed significantly increased splicing (decreased PSI) in the tumor, and INT_down_ was the number of minor introns that showed significantly decreased splicing (increased PSI) in the tumor. Then for each gene, minor splicing scores were correlated with average log_2_ fold change (Tumor/Normal) using Spearman’s rank test.

## 3. Results

### 3.1. Minor Intron-Containing Genes Tend to Be Upregulated in Tumors

Given the functional enrichment of minor introns in DNA processing genes, we predicted that the expression of minor intron-containing genes (MIGs) would be upregulated in tumor tissue. Across 14 different TCGA cohorts, we analyzed differential expression of 623 MIGs between matched tumor and adjacent normal tissue (Figure 1). In all cohorts analyzed, MIGs tended to be upregulated in the tumor compared to all protein-coding genes, as predicted (Figure 2; one sided binomial test, FDR < 0.05, corrected for number of cohorts tested).

### 3.2. TCGA Cohorts Show Diverse Patterns of Minor Splicing

Given that it has been shown that retention of minor introns can decrease transcript levels of host genes, we hypothesized that increased expression of MIGs in cancer could be due to increased splicing efficiency of minor introns [12]. To that end, we compared PSI for 110 minor introns between matched tumor and adjacent normal tissue across TCGA cohorts (Figure 1); however, we found no clear signal of increased splicing efficiency in cancers, with the splicing of minor introns showing very different patterns between cohorts (Figure 3). We found that in 3 cohorts (BRCA, THCA, and LUAD), there was a significant skew of minor introns showing increased splicing in the tumor (two-sided binomial test, FDR < 0.05, corrected for number of cohorts tested); while in 4 other cohorts (KICH, STAD, KIRC, and COAD), minor introns were significantly more likely to have decreased splicing in the tumor (two-sided binomial test, *p*-value < 0.05, corrected for number of cohorts tested). The other 7 cohorts (UCEC, HNSC, BLCA, KIRP, LUSC, PRAD, and LIHC) showed no significant directional trend of minor splicing.

Given the very different patterns across different cohorts, we used an ad hoc statistic called ‘minor splicing score’, to capture this heterogeneity. We defined the minor splicing score as the number of introns whose splicing is significantly increased in tumors minus the number whose splicing is significantly decreased. This feature well illustrates the clear differences: some cohorts have strong positive scores, some have strong negative ones, and others have a score close to zero. This score allowed us to further investigate patterns of general increased or decreased minor splicing activity across cohorts. Investigation of individual introns are beyond the scope of this study; however, all data are listed in Appendix A for further scrutiny.

### 3.3. Mutations in Minor Intron-Containing Genes Do Not Correlate with Change in Minor Splicing

One possible explanation for associations of changes in minor splicing with cancer is not general activity, but rather the effect of splicing efficiency differences on a single MIG. If so, it would not be surprising if that gene was shown to be highly mutated in the same cancers where minor splicing changed. To determine if this was the case, we compared cohorts based on percent of cases with a mutation (with no regard to the specific location of mutations within the gene—i.e., we identified highly mutated genes, not highly mutated introns) in 18 different MIGs that are frequently mutated in cancers (Appendix A). From this analysis, we did not identify any MIG which showed a similar mutation pattern to the minor splicing pattern (e.g., a MIG is shown to be frequently mutated in the high minor splicing cancers, but not the others).

### 3.4. No Clear Patterns of Minor Splicesome Expression

Given the very different minor splicing patterns observed across cohorts, we performed several analyses to identify potential regulatory mechanisms that would explain these differences. We first looked at differential expression (matched tumor vs. normal tissue) of 15 known minor spliceosomal components (11 protein-coding genes and 4 snRNAs) across all cohorts, with the prediction of general upregulation in the high minor splicing cohorts and downregulation in the low minor splicing cohorts (Appendix A). Within a given cohort, we did not see any general directional change of the components. Furthermore, when looking across cohorts, we did not find any spliceosomal component whose change in expression correlated with changes in minor splicing (e.g., upregulated in high minor splicing cancers, but downregulated in low minor splicing cancers). One surprising finding was that all four minor spliceosome snRNAs were strongly downregulated in the thyroid cancer cohort (THCA), which showed increased minor splicing.

Given these surprising patterns of spliceosomal component expression across cohorts, we wanted to dig deeper and look at correlation of the component expression with minor splicing activity within individual cohorts. To this end, minor splicing activity was estimated for all tumors based on PSI values for all 110 minor introns, which was then correlated with component expression (Appendix A). Surprisingly, 5 minor spliceosomal components (ZRSR2, SNRNP48, SNRNP35, RNU4ATAC, and RNPC3) showed a significant negative correlation with minor splicing across most if not all cohorts (14/14 for ZRSR2, 9/14 for SNRNP48, 14/14 for RNU4ATAC, and 12/14 for RNPC3). The only component to show a fairly consistent positive correlation with minor splicing activity was ZCRB1 (9 out of 14 cohorts). We also correlated changes in minor splicing activity with changes in component expression between matched tumor and adjacent normal tissue (Appendix A). In that analysis, RNPC3 again showed a relatively consistent negative correlation with minor splicing activity (6 out of 14 cohorts). In the BRCA cohort (which had the largest number of patients and thus the greatest statistical power), 5 components (ZMAT5, ZCRB1, SNRNP35, SNRNP25, and ARMC7) had a significant positive correlation with minor splicing.

### 3.5. Patterns in TCGA Cohorts Are Not Associated with a Particular Regulatory Netowrk

Since expression of the minor spliceosome components did not clearly distinguish the high minor splicing cohorts from the low minor splicing cohorts, we began looking at genes not related to minor splicing for insight into the different patterns across diseases. Given that minor splicing was shown to be high in stem cells, we hypothesized that the high minor splicing cohorts might have more stemness than the low minor splicing cohorts. To that end, we analyzed differential expression between matched tumor and adjacent normal tissue for 48 genes from a defined human adult stem cell gene signature [28]; however, we did not see any trends that explained the different minor splicing patterns between cohorts (Appendix A). Furthermore, we analyzed the functional enrichment of the upregulated and downregulated MIGs (Appendix A) and the MIGS that showed increased and decreased splicing (Appendix A) in the cohorts. While there was some enrichment in some cohorts for differentially expressed MIGs, there was no functional enrichment of MIGs that showed differential intron retention. Finally, we correlated changes in gene expression of each protein-coding gene with changes in minor splicing across cohorts (Appendix A). From this, the uncharacterized C7orf26 was the only gene that showed significant correlation with changes in minor splicing across cohorts (Spearman R = −0.957, FDR = 0.0016).

## 4. Discussion

We report complex patterns of minor intron splicing across 14 cohorts of The Cancer Genome Atlas. While minor intron-containing genes showed a significant tendency to be upregulated across all tumors, changes in minor splicing had pronounced variation among cohorts. Minor intron retention was shown to increase, decrease, or show no evident change depending on the cohorts analyzed.

The diversity of minor splicing patterns across cancers is, in some ways, distinct from patterns found in overall intron retention across all introns. Whereas we found strikingly opposite patterns across cancers, Dvinge and Bradley found in all introns that retention was higher in all cancers, except breast cancer, for which further analysis revealed relatively high intron retention in normal breast tissue [30]. While the breast cancer cohort exhibits the most striking elevation in minor splicing, it was not the only cohort to show such a skew. In both lung adenocarcinoma and thyroid cancer, minor introns were more likely to have increased splicing in the tumor than decreased. This would indicate that the mechanisms that regulate minor splicing in cancer are distinct from those that regulate overall splicing.

The prevalence of negative correlations between minor spliceosomal component expression and minor splicing activity was also quite surprising. A possible explanation for this contradictory relationship is negative autoregulatory feedback loops, where component transcript levels are lower when minor splicing activity is high. This was shown to be the case for both RNPC3 and SNRNP48, where minor splicing in RNPC3 leads to a longer, destabilizing 3′ UTR; while minor splicing in SNRNP48 leads to inclusion of a non-constitutive exon, which causes NMD [31]. Other components may experience similar but more complex feedback loops, where one of their upstream regulators is affected by minor splicing. These findings bring into question the appropriateness of using component expression as an indicator of minor splicing activity.

Limitations to this study include relying on RNA-seq data from bulk tumor tissue, which often include non-cancerous cells such as immune cells, and those that make up the vasculature. It is possible the trends we observed are in part due to the presence of these cells, rather than the cancer cells themselves. Furthermore, there could be minor introns that are important to tumorigenesis but are in genes that are very lowly expressed; thus, there would be too few reads to have sufficient statistical power to detect its importance.

## 5. Conclusions

Our results stand in striking contrast to clear hypotheses for directional effects of minor intron splicing in cancer, whether positive due to the association between minor splicing and cell cycle progression, or negative due to loss of function of minor intron-containing tumor suppressor genes. The unexpected patterns reported could possibly simply reflect the differential impact of these two directional forces; alternatively, positive and negative effects on splicing in different cancers could be a pleiotropic effect of other changes that are correlative rather than causative of cancer progression. These results provide a resource for further explorations of the context-dependent roles of minor introns in cancer, which could lead to development of novel therapies, including antisense oligonucleotides, small molecules, and immune activating strategies.

## Figures and Tables

**Figure 1 genes-13-00387-f001:**
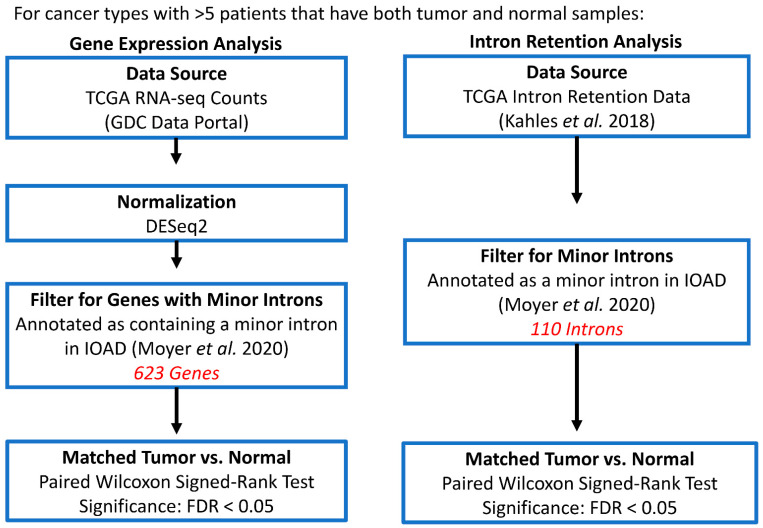
Flowchart of the computational pipeline to identify differentially expressed genes with minor introns (**left** panel) and spliced minor introns (**right** panel) between matched tumor and adjacent normal tissue (see details in Section 2).

**Figure 2 genes-13-00387-f002:**
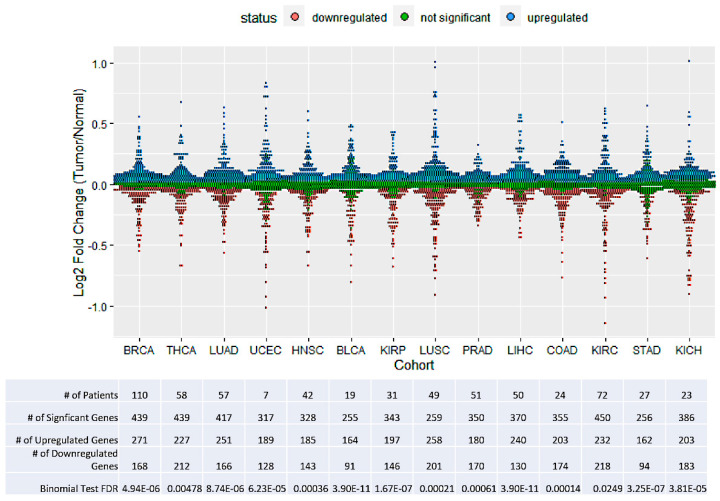
Minor intron-containing genes are generally upregulated in cancer. Expression fold change between tumor and adjacent normal tissue of minor intron-containing genes. Within each cohort, each dot represents the average log_2_ fold change (tumor/normal) of a minor intron-containing gene within the given cohort. Paired Wilcoxon signed-rank test was performed to determine significance, FDR < 0.05. For each cohort, there were more upregulated minor intron-containing genes than would be expected compared to all protein-coding genes (one sided binomial test, FDR < 0.05, corrected for number of cohorts tested). Colors indicate significantly upregulated and downregulated genes as well as genes without significant differences (blue, red, and green, respectively), thus the preponderance of blue over red dots demonstrates the preponderance of minor intron-containing genes to have increased expression in tumors. These data are also summarized in the corresponding table below, showing the number of genes that were significantly upregulated or downregulated (equal to the numbers of blue and red dots).

**Figure 3 genes-13-00387-f003:**
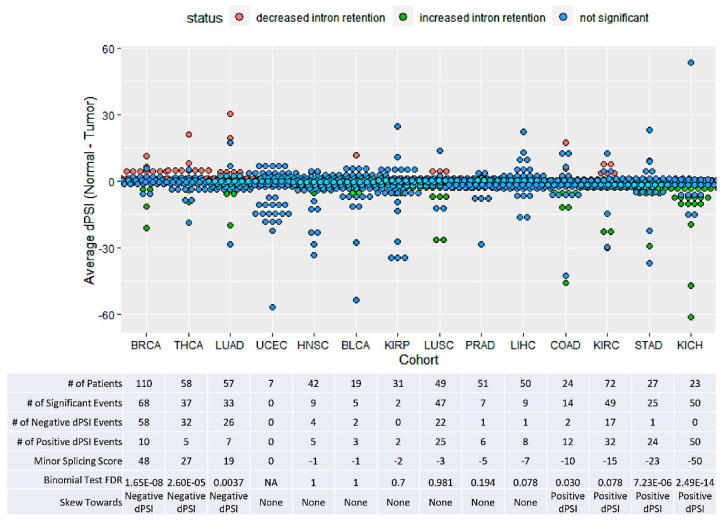
Minor splicing patterns vary between TCGA cohorts. Differential splicing between tumor and adjacent normal tissue of minor introns. Each dot represents the average dPSI of an intron (normal–tumor) within the given cohort. Paired Wilcoxon signed-rank test was performed to determine significance, FDR < 0.05. A two-sided binomial test was performed to determine if the significant events skewed towards one side (FDR < 0.05, corrected for the number of cohorts tested). As in Figure 2, blue and red dots represent significant differences and green dots had non-significant differences, and the table summarizes these results Cohorts are ordered by their minor splicing score (# of negative dPSI events to # of positive dPSI events).

## Data Availability

Access to the data presented in this study is available in the Section 2 and in the Appendix A section.

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
