# Peer review of "Distinct Minor Splicing Patterns across Cancers"

_genes, 2022, doi:10.3390/genes13020387_

Round 1

Reviewer 1 Report

Levesque and colleagues, here provide an analysis of the dysfunction of a not well understood biological component across human cancers: the minor spliceosome. Through an in silico analysis starting from publicly available data, the authors show the patterns of minor intron splicing across pancancer studies, highlighting among other points that genes containing minor introns tend to be upregulated in tumors. However, because of the great heterogeneity in minor intron retention, it remains difficult to identify specific dysfunctional phenotypes. Overall, the authors make a big effort to try to maintain a rational flow across the manuscript with an analytical approach that is not among the easiest to understand. The paper is overall well written, although the results shown unfirm almost every time the initial hypothesis. Even if this could be considered a drawback for this work, I agree that is important to show negative findings. However, the authors should make some efforts to make the paper better understandable for a broader reading community.

I would suggest better introducing the mechanisms of splicing, their importance for physiological and pathophysiological mechanisms before going ahead with the unconventional splicing.

Then I advise to better elaborating on why it should be important to analyze minor splicing and which is the difference between U12 and U2 introns. A description of the cellular components involved would be welcome.

Then from an analytical point of view, it is not 100% clear whether the metrics related to the computation of minor splicing were already calculated in a previous study or not. In this case, the previous work should be described and a clear acknowledgment in the manuscript of how this paper would advance the knowledge on this topic should be made.

I would suggest providing a study design flowchart to make clear all the steps of the study and the working hypotheses. Also, it is hard to interpret Figure 1 and 2. I understand that each dot represents the average logFC of tumor vs normal expression of the genes containing minor introns across each cohort. I think that these figures do not drive the main message that seems to be (in figure 1) that most tumor tissues have minor intron-containing genes upregulated compared to normal tissues. Could it be possible to figure out another visualization capable to highlight the main findings? Another useful analysis with annexed visualization could be to show the gene pathways enriched in minor intron-containing genes in normal and tumor tissues.

Author Response

We thank Reviewer 1 for their thoughtful feedback and suggestions. Below we have addressed each comment specifically, with reviewer comments in bold, while our responses are not in bold.

1. I would suggest better introducing the mechanisms of splicing, their importance for physiological and pathophysiological mechanisms before going ahead with the unconventional splicing.
We have added an introductory paragraph (lines 24-32) to introduce the concept and importance of splicing to an audience that might be unfamiliar with it. We briefly explain the basic molecular process that is taking place, followed by why it is an important molecular and physiological process. We also provide examples of how splicing is regulated in biological processes.

2. I advise to better elaborating on why it should be important to analyze minor splicing, and which is the difference between U12 and U2 introns. A description of the cellular components involved would be welcome.
We agree that it is important to emphasize how U2 and U12 introns differ. We have added a section detailing how the U2 and U12 spliceosomes differ in their composition (lines 46-51). In addition, we have reworded the last sentence in paragraph 2 (line 43), and the sentence where we discuss the sequence difference in U2 and U12 introns (lines 51-54), to emphasize that U2 and U12 introns are spliced by similar, but different machineries.

3. In this case, the previous work should be described and a clear acknowledgment in the manuscript of how this paper would advance the knowledge on this topic should be made. I would suggest providing a study design flowchart to make clear all the steps of the study and the working hypotheses.
We have added a flowchart (Figure 1) that details the computational pipeline to identify differentially expressed genes with minor introns (left panel) and spliced minor introns (right panel). In that flowchart, we have also cited where the source data has come from. The flowchart is referenced it in lines 152 and 181.

4. Also, it is hard to interpret Figure 1 and 2. I understand that each dot represents the average logFC of tumor vs normal expression of the genes containing minor introns across each cohort. I think that these figures do not drive the main message that seems to be (in figure 1) that most tumor tissues have minor intron-containing genes upregulated compared to normal tissues. Could it be possible to figure out another visualization capable to highlight the main findings?
While we do believe this is the best way to visualize the data, we have added a more thorough explanation of the data to the figure legends (previous figure 1, now figure 2, lines 163-174; previous figure 2, now figure 3, lines 200-207).

5. Another useful analysis with annexed visualization could be to show the gene pathways enriched in minor intron-containing genes in normal and tumor tissues.
We did an additional analysis of functional enrichment of the upregulated and downregulated MIGs (Supplemental Table 2) and the MIGS that showed increased and decreased splicing (Supplemental Table 3) in the cohorts. While there was some enrichment in some cohorts for differentially expressed MIGs, there was no functional enrichment of MIGs that showed differential intron retention. Methods are detailed in lines 136-138, and results are described in lines 261-266.  In addition, the title of section 3.5 (lines 250-252) has been changed to better represent the new information.

Reviewer 2 Report

The manuscript provides a comprehensive approach to a very interesting and outstanding question in the field of splicing and cancer. Many reports point to the involvement of minor intron splicing in cancer but not much is done to address it. While the first part of the manuscript starts off in the right direction, the authors opted to keep their analysis somehow superficial. To be more specific, one reason the authors did not find correlations between minor intron splicing and almost any of the variables they looked at is the fact that they are thinking of minor introns and MIGs as one homogeneous class of genes. In fact, besides the highly conserved splice sites, very little about the features of these genes is homogeneous, which makes it hard to easily find patterns and/or correlations using simple statistical methods, like the ones used in this manuscript. One have to use more sophisticated approaches and even maybe AI and machine learning approaches to solve this problem. 

Some of the specific comments:

  • the authors should address if there are any differences in the features/functions of minor introns whose splicing is up vs. down in any of these cohorts. Take BRCA cohort for example, what is the difference (functional or otherwise) between the 271 upregulated genes and the 168 downregulated genes? Then what is the difference between the 58 with -ve PSI and the 10 with +ve PSI value?
  • in section 3.3, did the authors look at any mutation anywhere in the gene or did they focus on mutation in or close to the minor intron? It is important to make this distinction. 

Author Response

We thank Reviewer 2 for their thoughtful feedback and suggestions. Below we have addressed each comment specifically, with reviewer comments in bold, while our responses are not in bold.

1. To be more specific, one reason the authors did not find correlations between minor intron splicing and almost any of the variables they looked at is the fact that they are thinking of minor introns and MIGs as one homogeneous class of genes. In fact, besides the highly conserved splice sites, very little about the features of these genes is homogeneous, which makes it hard to easily find patterns and/or correlations using simple statistical methods, like the ones used in this manuscript.
We agree with the reviewer that minor intron containing genes are not a homogeneous class and vary in their functions as well as expression patterns. The reason we analyzed splicing of all minor introns, was to determine if there was evidence of changes in the global activity of the minor splicing machinery during tumorigenesis, as assessed by intron retention. We have added an additional sentence at the end of the introductory section (lines 81-84) to clarify the rationale for our methods and what question we are investigating.

2. The authors should address if there are any differences in the features/functions of minor introns whose splicing is up vs. down in any of these cohorts. Take BRCA cohort for example, what is the difference (functional or otherwise) between the 271 upregulated genes and the 168 downregulated genes? Then what is the difference between the 58 with -ve PSI and the 10 with +ve PSI value?
We did an additional analysis of functional enrichment of the upregulated and downregulated MIGs (Supplemental Table 2) and the MIGS that showed increased and decreased splicing (Supplemental Table 3) in the cohorts. While there was some enrichment in some cohorts for differentially expressed MIGs, there was no functional enrichment of MIGs that showed differential intron retention. Methods are detailed in lines 136-138, and results are described in lines 261-266.  In addition, the title of section 3.5 (lines 250-252) has been changed to better represent the new information.

3. In section 3.3, did the authors look at any mutation anywhere in the gene or did they focus on mutation in or close to the minor intron? It is important to make this distinction.
We have clarified the details of the mutation analysis in lines 214-216.

Round 2

Reviewer 2 Report

We thank the authors for clarifying some of the earlier points of concern and editing the manuscript accordingly

All of this reviewer's comments were satisfactorily addressed.